# Autonomous Scheduling of Agile Spacecraft Constellations with Delay Tolerant Networking for Reactive Imaging

**Sreeja Nag[1], Alan S. Li[1], Vinay Ravindra[1], Marc Sanchez Net[2], Kar-Ming Cheung[2], Rod Lammers[3]**, and **Brian Bledsoe[3]**

[1]NASA Ames Research Center, Bay Area Environmental Research Institute, CA, USA
[2]Jet Propulsion Laboratory, California Institute of Technology, CA, USA
[3]University of Georgia, Athens GA, USA
sreejanag@alum.mit.edu

## Abstract

Small spacecraft now have precise attitude control systems available commercially, allowing them to slew in 3 degrees of freedom, and capture images within short notice. When combined with appropriate software, this agility can significantly increase response rate, revisit time and coverage. In prior work, we have demonstrated an algorithmic framework that combines orbital mechanics, attitude control and scheduling optimization to plan the time-varying, full-body orientation of agile, small spacecraft in a constellation. The proposed schedule optimization would run at the ground station autonomously, and the resultant schedules uplinked to the spacecraft for execution. The algorithm is generalizable over small steerable spacecraft, control capability, sensor specs, imaging requirements, and regions of interest. In this article, we modify the algorithm to run onboard small spacecraft, such that the constellation can make time-sensitive decisions to slew and capture images autonomously, without ground control. We have developed a communication module based on Delay/Disruption Tolerant Networking (DTN) for onboard data management and routing among the satellites, which will work in conjunction with the other modules to optimize the schedule of agile communication and steering. We then apply this preliminary framework on representative constellations to simulate targeted measurements of episodic precipitation events and subsequent urban floods. The command and control efficiency of our agile algorithm is compared to non-agile (11.3x improvement) and non-DTN (21% improvement) constellations.

## Introduction

Response and revisit requirements for Earth Observation (EO) vary significantly by application, ranging from less than an hour to monitor disasters, to daily for meteorology, to weekly for land cover monitoring (Sandau, Roeser, and Valenzuela 2010). Geostationary satellites provide frequent revisits, but at the cost of coarse spatial resolution, extra launch costs and no polar access. Lower Earth Orbit satellites overcome these shortcomings, but need numbers and coordination to make up for response characteristics. Adding agility to satellites and autonomy to the constellation improves the revisit/response for the same number of satellites in given orbits. However, human operators are expected to scale linearly with constellation nodes (Eickhoff 2011) and operations staffing may be very costly.

### Earth-Observing Constellation Autonomy

Scheduling algorithms for agile EO have been successfully developed for single large satellite missions, examples being ASPEN for EO-1, scheduling for the ASTER Radiometer on Terra, high resolution imagery from the IKONOS commercial satellite (Martin 2002), scheduling observations for the geostationary GEO-CAPE satellite (Frank, Do, and Tran 2016), scheduling image strips over Taiwan by ROCSAT-II (Lin et al. 2005), and step-and-stare approaches using matrix imagers (Shao et al. 2018). The Proba spacecraft demonstrated dynamic pointing for multi-angle imaging of specific ground spots that it is commanded to observe (Barnsley et al. 2004). Scheduling 3-DOF observations for large satellite constellations has been formulated for the PLEIADES project (Lemaître et al. 2002; Damiani, Verfaillie, and Charmeau 2005) and COSMO-SkyMed constellation of synthetic aperture radars (Bianchessi and Righini 2008). Scheduling simulations have demonstrated single Cubesat downlink to a network of ground stations within available storage, energy and time constraints. (Chien et al. 2019) has developed automated tasking for current sensors as a Sensor Web to monitor Thai floods.

Recent advances in small and agile satellite technology have spurred literature on scheduling fleet operations. Coordinated planners in simulation (Abramson et al. 2013; Robinson et al. 2017) can handle a continuous stream of image requests from users, by finding opportunities of collection and scheduling air or space assets. Cubesat constellation studies (Cahoy and Kennedy 2017) have successfully scheduled downlink for a fleet, aided by inter-sat communication. Evolutionary algorithms for single spacecraft (Xhafa et al. 2012), multiple payloads (Jian and Cheng 2008) and

satellite fleets (Globus et al. 2002) are very accurate but at large computational cost due to their sensitivity to initial condition dependence (genetic algorithms), exponential time to converge (simulated annealing) or large training sets (neural nets). Agile constellation scheduling with slew-time variations have shown reasonable convergence in the recent past using hierarchical division of assignment (He et al. 2019). However, algorithms have not been developed for onboard execution on real-time, fast-response EO applications and do not consider inter-sat comm scheduling in conjunction with imaging operations.

We have recently demonstrated (Nag, Li, and Merrick 2018) a ground-based, autonomous scheduling algorithm that optimizes spacecraft attitude control systems (ACS) to maximize collected images and/or imaging time, given a known constellation. The algorithm is now broadened in application scope by leveraging inter-satellite links and onboard processing of images for intelligent decision-making. Improving coordination among multiple spacecraft allows for faster response to changing environments, at the cost of increased scheduling complexity.

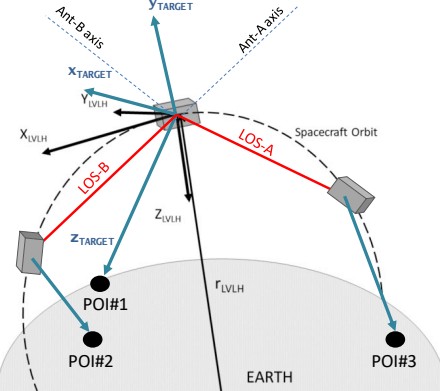

Figure 1—A constellation of satellites observing three points of interest (POI) by agile steering of their body frames, based on information shared when they have line of sight (LOS).

## Networked Constellations and Reactive Science
DARPA's delay/disruption tolerant network or DTN paradigm (Cerf et al. 2007) is an emerging protocol standard for routing in the dynamic and intermittent operation environment. DTN makes it possible to minimize replication and improve the delivery probability within available resources, but has never been applied to EO inter-sat data exchange. We show that DTN enabled, agile constellations can respond to transient, episodic and/or extreme events using an autonomous scheduling algorithm that is executable onboard. The target scenarios are simulated by a simplified Observing System Simulation Experiment (OSSE) to evaluate the benefit of our proposed algorithm.
In the traditional sense, an OSSE is a data analysis experiment used to evaluate the impact of new observing systems,

e.g. satellite instruments, on operational forecasts when actual observational data are not fully available (Arnold and Dey, 1986). An OSSE comprises of a free-running model used as the ground truth ('nature run'), used to compute the 'synthetic observations' for any observing systems, with added random errors representative of measurement uncertainty. Synthetic observations represent a small, noisy subset of the ground truth. They are used to forecast the full ground truth, then compared with the nature run. The disparity between the nature run of a chosen scenario, and the instrument-derived forecasts is then used to inform better instrument or mission design (Feldman et al. 2011). OSSEs can be used to train heuristics for mission planning, because different operational options can be assessed for different relevancy scenarios by changing the observing system characteristics and nature run appropriately (Nag, Gatebe, and Weck 2015; Nag et al. 2016).

## Methodology
We propose a novel algorithmic framework that combines physical models of orbital mechanics (OM), attitude control systems (ACS), and inter-satellite links (ISL) and optimizes the schedule for any satellite in a constellation to observe a known set of ground regions with rapidly changing parameters and observation requirements. The proposed algorithm can run on the satellites, so that each can make observation decisions based on information available from all other satellites, with as low a latency as ISL allows. Satellites generate data bundles after executing scheduled observations to be broadcast by ISLs. Bundles contain information about the ground points observed and meta-data parameters pre-determined by the OSSE. Considering networking delays in a temporally varying disjoint graph (e.g. results in Figure 5) and diminishing returns for observing fast-changing environments, satellites are not expected to iterate on acknowledgments to establish explicit consensus. Instead, the more a satellite knows about a region before its observation opportunity, better its scheduler performance. The algorithm may also run on the ground, i.e. the satellites can downlink their observed data, the ground will run the proposed algorithms, and uplink the resultant schedule to the satellite. Since the ground stations are expected to be inter-connected on Earth and in sync with each other at all times, the optimization is centralized and the resultant schedule avoids potentially redundant observations due to lack of consensus among the satellites. This approach also reduces the onboard processing requirements on the satellites. However, since information relay occurs at only sat-ground contacts (function of orbits, ground network), the scheduler would use significantly information compared to the distributed, onboard bundles. The transiency of the environment being observed and its robustness to latency in exchanging inferences determines effectiveness of the onboard, decentralized vs.

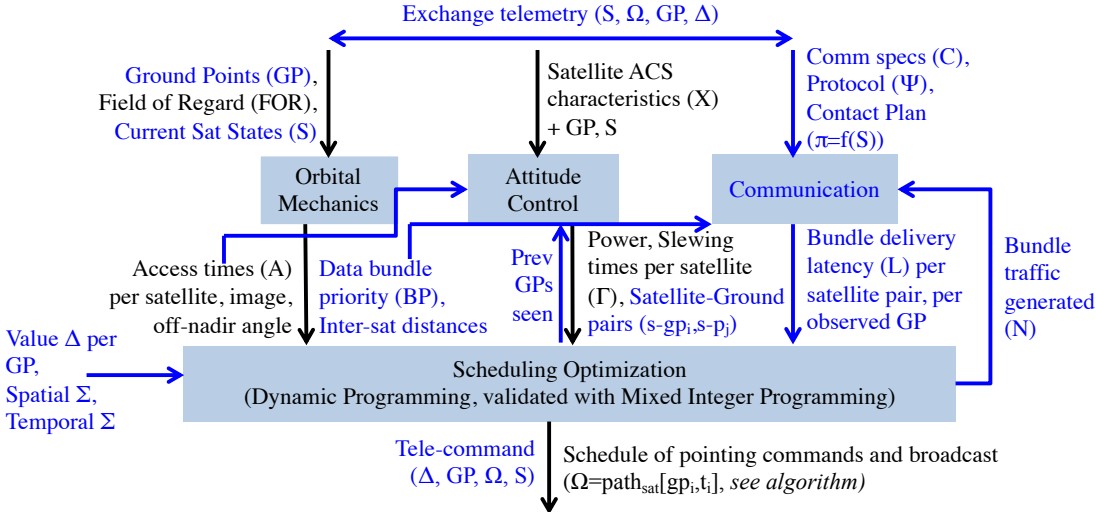

Figure 2 – Major information flows between the modules in the proposed agile EO scheduler, expected to run onboard every satellite in a given constellation, applied to global urban flooding in this paper. This framework can exchange information (as identified at the top) between the satellites via peer-to-peer communication or via the ground (reverse bent pipe architecture). The blue arrows/text represent newly added components to the previous version of the algorithmic framework (Nag, Li, and Merrick 2018).

ground, centralized implementation of our proposed algorithm (e.g. scenario results in Table 2). The algorithmic framework and information flow summarized in Figure 2.

The OM module propagates orbits in a known constellation and computes possible coverage of known regions of interest (appropriately discretized into ground points – GP) within a field of regard (FOR). It provides the propagated states and possible access times per satellite, per GP to the ACS. The ACS uses this information, with known satellite and subsystem characteristics to compute: time required by any satellite at a given time to slew from one GP to another (including satellite movement), resultant power, momentum and stabilization profiles. The OM also provides available line-of-sight (LOS) availability, corresponding inter-sat distances at any time, as well as the priority of bundle delivery to the Comm/ISL module so that it knows which satellites need to receive the data sooner. The comm. module computes the link budget for a known set of specifications and protocols, and uses the resultant data rate to simulate DTN and compute bundle drop rates and latency to deliver any known bundle between any given pair of satellites. Bundles exchanged between the satellites are modeled to contain seen GPs time series ($\Omega$) or new GPs of interest, and their re-computed value ($\Delta$), either in full, an update to the original, or as parametric meta-data. The mechanism of re-computing value at a GP (input on the left of Figure 1) is described on pg.4.

The optimization module ingests the outputs of the OM, ACS and Comm modules to compute the schedule of when each satellite should capture any GP. The executed schedule dictates the number of observations a satellite will make

over any region, which dictates the number, size and timing of bundles generated for broadcast, therefore, we include a feedback loop between the optimizer and the comm. module. Slew characteristics depend on the previous GPs the satellite was observing and intended next, thus a feedback loop between the ACS and optimizer. If the constellation specifications, e.g. number of satellites, their arrangement, instruments, FOR, are expected to change over operational lifetime, a feedback between the OM and the optimizer may also be added. In the current implementation, we assume that the proposed real-time scheduling will take a fixed time interval and that other operations, e.g. downlink, calibration, maintenance, etc., will be scheduled separately.

**Orbital Mechanics and Attitude Control**

The main revision to the OM and ACS modules comprehensively described in (Nag, Li, and Merrick 2018) is that we now compute slewing time as a function of a pair of $sat_{t,s}$-$gp_i$, each representing a vector from satellite $s$ at time $t$ to ground point $i$. The dynamic programming (DP) algorithm in the optimizer now uses observable GPs as potential states, instead of discrete pointing options. Since the DP scheduler iteratively calls ACS for any pair of vectors, the ACS slew time cannot be pulled from a static table using the starting and ending satellite pointing direction as before, and are computed real-time. Onboard processing constraints limit our use of the full physics-based ACS simulator (developed on MATLAB Simulink), therefore we developed weighted least squares on a third order polynomial whose coefficients are a function of the satellite mass, ACS and other specifications that served as knobs in the original model. The polynomial provides a very efficient implementation of the

ACS-DP feedback loop in Figure 2, by allowing fast computation of slew time as a function of α, the angle between any pair of vectors $sat_{t,s}$-$gp_i$. This process is adaptable to non-planar angle dependencies as well, in case a full body re-orientation of the satellite is necessary, not just re-pointing an instrument.

The OM module has been developed in house, leveraging NASA GSFC's open-source General Mission Analysis Tool (Hughes 2007). The OM also generates data bundle priority, to be ingested by the comm. module as follows: The data-bundle is tagged with the corresponding region or point of observation (where the data is generated). Priority is given to the next satellite to be able to access the same region or point, after the bundle generation, so that when it reaches the said region, it is up-to-date about it, as inferred by the last observing satellite. For example, if Sat11 generates data over "Dallas", and Sat12 is the next satellite on scene, Sat12 is given highest priority for the data-bundle to be delivered. If some satellites do not ever visit "Dallas", they are removed from the recipient queue.

### Delay/Disruption Tolerant Networking

Delay/Disruption Tolerant Networking (DTN) is a new set of communication protocols developed with the intent of extending the Internet to users that experience long delays and/or unexpected disruptions in their link service. At its core, DTN defines an end-to-end overlay layer, termed "bundle layer", that sits on top of the transport layer (i.e., TCP or UDP in the Internet, CCSDS link layer standards in space) and efficiently bridges networks that may experience different types of operational environments. To achieve that, it encapsulates underlying data units (e.g., TCP/UDP datagrams) in bundles that are then transmitted from bundle agent to bundle agent, who store them persistently for safe-keeping until the next network hop can be provisioned. This hop-to-hop philosophy is at the core of DTN and differentiates it from the Internet, where transactions typically occur by establishing end-to-end sessions (between the data originator and data sink).

At present, DTN is comprised of a large suite of specifications that encompass all aspects of network engineering, including its core protocols (e.g., the Bundle Protocol (Scott and Burleigh 2007), the Licklider Transmission Protocol (Farrell, Ramadas, and Burleigh 2008), or the Schedule Aware Bundle Routing (Standard and Book 2018)), adapters for bridging different types of underlying networks, network security protocols and network management functionality (Asynchronous Management Protocol). For the purposes of this paper, however, only the parts of the Bundle Protocol and Schedule Aware Bundle Routing Protocol were implemented. Together, they provide a medium to high fidelity

estimate of how bundles would move in a DTN consisting of near-Earth orbiting spacecraft and allow us to quantify network figures of merit such as bundle loss or average bundle latency. Our DTN model is implemented in Python using Simpy (Matloff 2008), a discrete-event engine built upon the concept of coroutines (or asynchronous functions in the latest Python versions).

### Quantifying Value: Observing System Simulations

We apply our proposed framework to episodic precipitation and resultant urban floods, to demonstrate its utility and scalability. We used the Dartmouth Flood Observatory (Brakenridge 2012) to study the frequency of global floods in 1985 – 2010, and identified 42 large cities that are within floodprone areas and marked a 100 km radius buffer around them to define the watersheds. We assume a 6 hour planning horizon, during which 5 of the 42 cities (Dhaka, Sydney, Dallas, London, Rio de Janeiro) flood to varying degrees as modeled by an OSSE nature run. For example, London tends to get slower, longer rains that might cause the Thames to flood, Dallas is more concerned with short, intense thunderstorms causing flooding on smaller creeks. The OSSE developed for this paper uses an area of 80km x 80km, and is currently agnostic to flood-type disparities between cities.

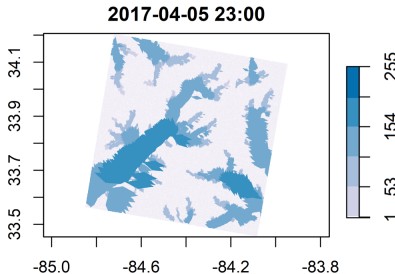

Figure 3 – Example of the spatial distribution of value (8-bit scale) of an ~80 km square region around Atlanta, GA (X/Y axis in degrees), for a single snapshot in time.

To quantify value for the ***optimizer's objective function***, we modeled riverine flooding in the Atlanta metropolitan area for a single storm event from April 5-6, 2017, using the WRF-Hydro hydrologic model version 5 (Gochis et al. 2018). The model was run with a grid resolution of 900 m, and was calibrated by adjusting parameters until modeled streamflow matched measured flow at nine U.S. Geologic Survey gages in the Atlanta metro area. The modeled channel flow rates were then normalized by the 2-year recurrence interval flow rate (Q2), as estimated from the USGS regional regression equations for urban streams in that region (Feaster, Gotvald, and Weaver 2014). Q2 is the flow that has a 50% chance of happening in any year, and is an estimate of what constitutes a "flood" at any location. Finally, these normalized flood rates were then transformed on a log-scale to integer values between 1 and 256. We estimated the

watershed land area draining to each channel point and the [1,256] value of that point was assigned to the entire watershed area. Areas with high value correspond to watersheds with active flooding. In these watersheds, it is important to obtain satellite-derived estimates of precipitation to determine if this flooding will worsen (with more rainfall) or abate. This process provides an expected value of observing every point in a region of interest at 15 min resolutions, to be used by the satellite scheduler, $absval([gp_x,t_y])$ in the next section. One snapshot is shown in Figure 3.

This paper uses the following statistical model for value re-computation. Since the OSSE time resolution is 15min, the cumulative value of observing any GP within 15min should be constant, i.e. if it has been seen once, subsequent observations within 15min are of zero value. Since value re-computation based on collected data is not physically simulated, we estimate it from OSSE output ($absval$) in the following ways: The value of observing any GP after 15min is considered a fraction ($=1/number\_of\_times\_seen$) of its OSSE-provided value with some random noise added. This re-computation ensures that a diversity of GPs is observed over time. Similarly, the value of observing a GP can be inversely proportionate to its distance to already observed GPs, to maximize the spatial spread of information collected and characterize the region better. The time-stepping nature of our algorithm causes it to be agnostic to the future value of any GP, therefore it can ingest changing values as they come along and compute schedules accordingly. We applied a standard normal distribution with a 2%-8% (uniformly random) standard deviation to the OSSE-provided values, to generate slightly different value functions to be used by each satellite's optimizer. This was to simulate different inferences by satellites after onboard processing their different observations, owing to different schedules. If they observed the same GPs at the same time, they would have the same inference, but that is obviously not possible. We are developing a high fidelity, value re-computation model to replace this statistical model, whose onboard processing algorithms and predictive technology will be described in a future publication. It will simulate processing data collected from executed observing schedules, updating future value, re-computing schedule and passing along insights the other sats.

### Dynamic Programming based Scheduler

Our proposed optimization algorithm (Table 1) uses dynamic programming (DP) to greedily optimize the scheduler. Each satellite is theorized to possess its own DP scheduler on board (or its own thread on ground) - a cartoon version is in Figure 4, where the gradient of the nodes represents varying $absval([gp_x,t_y])$, $\forall x \in [1,numGP]$, $y \in [1,horiz\_tSteps]$, as obtained from the OSSE. The scheduler outputs a vector of tuples $[gp_i,t_i]$ $\forall i \in [1,pathLength]$, which is the schedule for sat to observe $gp_i$ at $t_i$. Compared to our

previous implementation, the state space that the optimized path has to trace is now a graph of time steps and GPs, instead of time steps and satellite pointing directions. At any time $tPlan$ during mission operations, a schedule can be computed for a future $planning\_horizon$. The scheduler (line 4) processes bundles received until $tPlan$ through the DTN and updates its knowledge of all other sats $c$ as broadcasted at $tSrc_c$, i.e. $path_c[gp_i,t_i \leq tSrc_c]$, and its insights of the regions, i.e. $modelParams(t_i \leq tSrc_c)$. For every time step $tNow$, it then steps through the GPs $gpNow$ within the sat's FOR, and computes the cumulative value $val([gpNow,tNow])$ of each path ending at $[gpNow,tNow]$, e.g. in Figure 4. Via DP, line 11's computation entails adding $val([gpNow,tNow])$ to $val([gpBef,tBef])$ for all possible nodes $[gpBef,tBef]$ $\forall$ $gpBef \in [1,numGP]$, $tBef \in [$ $tNow-max(slewTime)$, $tNow-min(slewTime)]$. $slewTime$ is the full y-space of Eq.(2) for representative set of reorientations in the current mission scenario. The nodes between the red horizontal lines in Figure 4 are examples of $[gpBef,tBef]$; searching only a practical portion of the space (e.g., $t-2$ through $t-9$) mitigates some of the computational load that has been added due to the dynamic slew computation, instead of the previous static, slew time table. Note that $val$ is re-computed using $absval$ and the satellite's knowledge of the executed observations by the rest of the constellation and their insights. Cumulative value is computed statistically as:

$$computeValue() = \sum_{x=1}^{numGP} \sum_{y=1}^{horiz\_tSteps} val([gp_x, t_y])$$

(1)

A future scheduler will implement a higher fidelity cross-correlation function which extends the current OSSE. If the scheduling sat's FOR at $tNow$ overlaps with any other's FOR (line 9), it must $computeValue()$ for all possible paths by

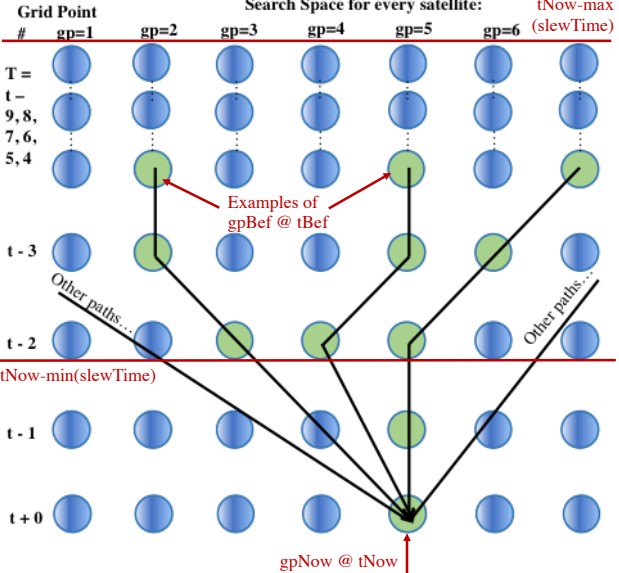

Figure 4—State space searched by the scheduler to compute the optimum path ending at $[gpNow,tNow]$, with no FOR overlaps

the other *s,* and maintain cumulative value numbers for possible paths by every permutation of sats in set *satsWoverlappingFOR* (line 13). Starting with the paths with maximum cumulative value, slew time of the last leg *[gpBef,tBef]→[gpNow,tNow]* (or combination of legs for overlapping sats) is dynamically computed. For the first instance where the time required is shorter than allowed between the *tBef→tNow* gap, the path is stored as the optimum path *path_sat[gpNow,tNow]*, and all other paths ending at *[gpNow,tNow]* discarded (to prevent memory becoming astronomical). Future implementations may explore ways to preserve some dominated paths since potentially optimum solutions, although tied with sub-optimum ones at *tNow*, are lost in the process.

Table 1—Summarized Scheduling Algorithm
1: Inputs – sat, absval([$gp_x$,$t_y$])
2: Output – $path_{sat}$[$gp_i$,$t_i$]
3: **For** c in Constellation-{sat} **do**
4:     modelParams($t_i{\leq}tSrc_c$), $path_c$[$gp_i$,$t_i{\leq}tSrc_c$] ←
            DTN(c,$tSrc_c$,sat,tPlan)
5: **End For**
6: **For** tNow in planning_horizon **do**
7:     **For** gpNow in GroundPntsInFOR(sat, tNow) **do**
8:         GroundPntsInBND=[tNow-max(slewTime):
                            tNow-min(slewTime),1:*numGP*]
9:         **For** s in satsWoverlappingFOR(sat,gpNow) **do**
10:            **For** **[**gpBef,tBef] in GroundPntsInBND **do**
11:                v[s]=computeValue($path_s$**[**$gpBef_s$,$tBef_s$]+
                    **[**gpNow,tNow],absval, $path_{c{\in}Const}$
                    [$gp_i$,$t_i{\leq}tSrc_c$], modelParams($t_i{\leq}tSrc_c$))
12:            **End For**
13:            v_combi = v[permute(s in satsWoverlappingFOR)]
14:            **For** vn in reverse_sort(v_combi)
15:                tslew=computeManueverTimes(s_combi,
                        [$gpBef_{s\_combi}$,$tBef_{s\_combi}$], [gpNow,tNow])
16:                **If** tslew≤[tNow-$tBef_{s\_combi}$] **then**
17:                    $path_s$[gpNow,tNow]←
                            $path_s$[$gpBef_s$,$tBef_s$]+[gpNow,tNow]
18:                    break // *forLoop for vn*
19:                **End If**
20:            **End For**
21:        **End For**
22:    **End for**
23: **End for**

The advantages of this algorithm are:
1. Runtime is linearly proportionate to the number of time steps in planning horizon n(T). The scheduler may be run for only the duration of FOR access over a region. It needs to be rerun only if value of the GPs (expected to be accessed) changes, as inferred or informed.
2. Since the scheduler steps through the planning horizon, it can be stopped at any given point in the runtime, and the resultant schedule is complete until that time step. Schedules can thus be executed as future schedules are being computed by the onboard processor.

3. A complex value function with non-linear dependencies (e.g. on viewing geometry or solar time) or multi-system interactions (e.g. Simulink or proprietary software calls) are easy to incorporate.
4. Algorithmic complexity per satellite per region to be scheduled is $O(n(T){\times}n(GP)^{2{\times}n(S)})$, where n(GP) is the number of ground points within FOR and n(S) is the number of satellites *that can access the same GPs at the same time*, i.e. GPs within FOR overlaps. Since GPs are typically designed to Nyquist sample the footprint, runtimes are instrument dependent. If satellite FORs are non-overlapping, runtime or space complexity does not depend on the size of the constellation. For well-spread constellations observing non-polar targets, n(S) = 1, or a couple.

Integer programming (IP) was able to verify that optimality of the above algorithm for single satellites was within 10%, and find up to 5% more optimal solutions (Nag, Li, and Merrick 2018). The DP solution was 22% lower than the IP optimality bounds for constellations, which is well within the optimality bounds of greedy scheduling for unconstrained submodular functions (Piacentini, Bernardini, and Beck 2019). The DP schedules were found at nearly four orders of magnitude faster than IP, therefore far more suited for real time implementations. Currently, the scheduler is (re)run at the same frequency as DTN-informed value (re)processing, however future implementations will explore methods to decouple them, because rapider value updates but longer planning horizon are better for solution quality.

## Results

We simulate a case study of 24 (20 kg cubic) satellites in a 3-plane Walker constellation observing floods in 5 global regions over a 6-hour planning horizon. All satellites are simulated at a 710 km altitude, 98.5 deg inclination, circular orbits similar to Landsat. The constellation is a homogeneous Walker Star-type, with 3 orbital planes of 8 satellites each. While the gap between satellite accesses to a region is ~10 mins when there is an orbital plane overhead, a minimum of 3 planes is needed for the maximum gap to be within 4.5 hours (median gap ~ 1hr). Two planes would not be able to appropriately respond to a 6-hr flood phenomenon even with agile pointing, crosslinks onboard autonomy. For the chosen altitude, at least 8 satellites per plane ensures consistent in-plane LOS (cross-plane LOS in polar regions only), therefore the 24-sat topology is the minimum nodes for continuous DTN to enable <6-hr urban flood monitoring.

Instruments potentially used for precipitation and soil moisture sensing are narrow field of view (FOV) radars, which justify the need to continuously re-orient the <10 km footprint to cover a large flooding area. Examples are the Ka-band radar on a cubesat called RainCube (Peral et al. 2015) for precipitation, L-band bistatic radar on CYGNSS, and a Cubesat P-band radar (Vega Cartagena et al. 2018) for soil moisture. This paper presents results for an 8km footprint

instrument. The field of regard (FOR) which limits the maximum off-nadir angle of the payload/instrument is set to 55 deg, because it corresponds to 5x distortion of the nadir ground resolution, which is the OSSE's limit to allow combining observations in a given region. Spatial resolution dependence of value can also be included in the objective function. The presented scenario will be varied in terms of the mission epoch and regions of interest since it affects access intervals, observations and bundle traffic, and performance sensitivity reported in a future publication.

The ACS model, characterized with the satellite specs from (Nag, Li, and Merrick 2018), is fitted by the following polynomial, where t is time for maneuver, and α is angle to span. The standard deviation is around 0.2116, so add 0.4232 to get ~95% percentile.

$$t = 6.1974 \times 10^{-6} \times \alpha^3 + 1.3904 \times 10^{-3} \times \alpha^2 + 1.4165 \times 10^{-1} \times \alpha + 4.6231$$

(2)

While we present results based on full body re-orientations of a small satellite, our proposed algorithm can support constraints from the gimbaled re-orientation of payloads for fixed, larger satellites, by replacing the *tslew* computation model in Table 1 line 15.

## Performance of Inter-Satellite Networking

To estimate the performance of the DTN protocol stack, we first evaluated the supportable data rate in the inter-satellite links between spacecraft in the constellation. We make the following assumptions: All spacecraft transmit at S-band within the 6MHz typically available to class A missions; the link distance is set to 6000km (from the OM module; we use the worst case for the inter-plane links since their distances are variable); they are equipped with an SSPA that can deliver up to 5W of RF power, and a dipole placed parallel to the nadir/zenith direction (typical for small sats). This design ensures minimal complexity since the SSPA can be directly connected to the antenna without needing splitters. Since the orientation of the spacecraft at any point in time is highly variable, we close the link budget assuming that both the transmitting and receiving antennas operate at the edge of the -3dB beamwidth. We consider that no atmospheric effects impair the links, and we select a ½ LDPC coding scheme together with a BPSK modulation, SRRC pulse shaping and NRZ baseband encoding. Using these inputs, we pessimistically estimate the link performance at 1kbps. Since multiple spacecraft can be in view of each other at any point in time (especially over the poles), and they carry omnidirectional antennas, there is potential for interference. For the physical layer, we assume that signal interference is mitigated using some form of multiple access scheme (e.g. Frequency or Code Division Multiplexing) – the reported 1kbps data rate must be interpreted as that presented by the

multiple access scheme to the upper layers of the protocol stack. Interference can also affect DTN's routing layer.

To route data through the time-varying topology of the 24 satellite constellation, we simulate the system assuming that each of them is a DTN-enabled node with a simplified version of the Bundle Protocol and the Schedule Aware Bundle Routing Protocol. The DTN simulation uses the following inputs: The OM-provided contact plan (opportunities between any satellite pair in the network) is the basis for all routing decisions, and is specified as a six element tuple: Start time, end time, origin, destination, average data rate, range in light seconds. Second, the traffic generated in the constellation, provided by the optimizer as a function of average collections, indicating when bundles are created, who sources them, who they are destined for, and the OM-provided relative priority flag with 14 levels. Bundles of size of 2000 bits (1645 bits of observational inference data plus 20% of overhead due to the protocol stack) are generated and broadcasted for every GP observation, and communications are assumed error-less at 1kbps. The priority levels are also used to set the Time-To-Live (TTL) property of all bundles such that: Priority 1 has a 15min TTL, priorities 2 and 3 have a 30min TTL, and priorities 4 to 15 have a 50min TTL. These rules let the network automatically discard stale information and minimize traffic congestion.

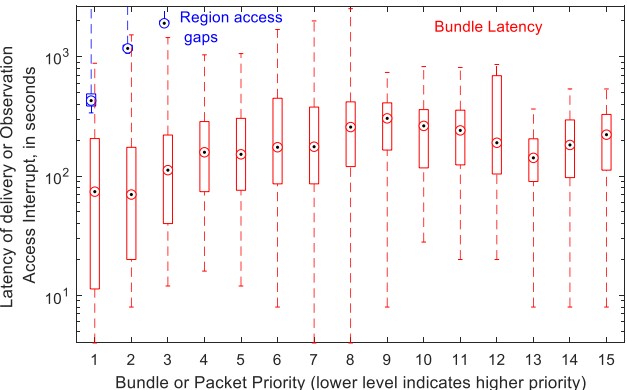

Figure 5 – Latency of data bundle delivery over all satellite pairs compared to the gaps between satellite FOR access to any region. For any satellite pair of given priority of DTN comm, if longest latency is less than shortest gap, each satellite can be considered fully updated with information the other, i.e. perfect consensus in spite of distributed scheduling on a disjoint graph. Each box represents 25%-75% quartiles, circle is median, whiskers show max/min

End-to-end latency experienced by 8341 bundles generated and sent over a 6 hour simulation (<1min DTN runtime) is shown in Figure 5. This latency is computed on a bundle-per-bundle basis, and measures the absolute time difference between the instant a bundle is delivered to the destination's endpoint (akin to TCP port), and the time it was originally created. Assuming a perfect multiple access scheme, any

spacecraft might receive a copy of a bundle that was not originally intended for it, causing the problem of packet duplication in the system due to physical interference. If not dealt with, these extra copies would be re-routed and create exponential replication problem that would overwhelm the entire system. To mitigate this, we take advantage of the extension blocks defined in the Bundle Protocol (Scott and Burleigh 2007). Particularly, every time router decides the next hop for a bundle, it appends an extension block with the identifier of the intended next hop. If another spacecraft receives a copy inadvertently, the router simply discards it.

Results indicate that latency is indeed affected by the bundle prioritization, however the effect is not monotonic because prioritization only happens at the bundle layer (e.g., radios have queues of frames, but they do not know about priorities in upper layers). Bundles with priorities 1-6 typically experience latencies of ~10s, with few outliers up to 15 minutes. This is quick enough for *most* of any satellite's knowledge to be transferred to the *next two* approaching any region (Figure 5). Also, no high priority bundles were dropped due to TTL expiration. Bundle with lower priorities experience larger latencies of ~2 minutes on average. The time to reach a region by satellites with priority≥3 is long enough for all bundles to be delivered, therefore all generated schedules by individual satellites have implicit consensus (because they use the same inputs). Access gaps for satellites with priority≥4 is out of Figure 5's Y-axis range. Latency was found to deteriorate non-linearly with increasing number satellites, bundle size due to more model parameters, and bundle traffic due to more observations. Future implementations will model bundle interference, trade-offs between omni vs. directional antennas, variations in bundle size and broadcast frequency, and their impact on latency.

## Performance of the Imaging Scheduler

The DP-based optimizer ingests outputs from all modules to find the best observational path that maximizes cumulative value till any given time, per satellite. We compare results from the use case in running the proposed algorithmic framework in 2 scenarios. *One,* the scheduler runs on onboard and uses collected information from other satellites as they come through the DTN every 10 minutes. Lowering (increasing) this re-scheduling frequency based on onboard power or processing constraints will improve (lower) the quality of results. *Two,* the scheduler runs on the ground and uses collected information from other satellites as they downlink. The ground stations are placed near both poles to emulate an optimistic scenario of ground contacts (thus, value update and rescheduling) twice an orbit i.e. ~30 per day. Lowering the contact frequency will lower the quality of results, e.g. current Cubesat missions commit to 2 contacts per day at NASA and 4-5 per day commercially.

The onboard run uses updated value of any GP, based on all bundles about that GP that have arrived (executed schedule and inference data from others ingested in Table 1 line 5). Since the scheduler is distributed and runs per satellite, it risks knowing everything or nothing about any GP, based on DTN's relay from other satellites. Our implementation shows that the constellation predicts GP value at an average of 4% different from their actual value, due to bundles about GPs arriving later than the satellite already observes them. This happens only for some outliers in the one or two hop connections (Figure 5), thus >95% of the GPs in all 5 regions are observed. Longer the DTN latency, more the difference between the assumed ("what it thinks it's seeing") and recorded value ("what it's actually seeing") of fast-changing phenomena, lower the cumulative value.

Table 2—Comparison of optimizers run Onboard vs. Ground. A constellation with no agility sees 8.4% of the GPs, in either case)

|  | Scenario#1 (Distributed) | Scenario#2 (Centralized) |
|---|---|---|
| Cumulative Value (6h) | 26347 | 21820 |
| % of all GP observed | 95.2% | 99.2% |

The centralized run has no risk of overlapping observations because all sats "know" every other's schedule, allowing for >99% of GPs seen. However, value functions are based on information obtained approximately an orbit earlier, due to collection-uplink-reschedule-downlink latency between any satellite pair. Our implementation shows that the constellation assumes GP value at an average of 70% different from recorded value, due to lack to timely communication of value updates. While the exact difference is a function of sensitivity of value updates to schedules executed by different sats (currently fractional decay with observation, 2%-8% variation in inference), it shows that in fast changing environments, a responsive constellation's performance is better captured by OSSE-driven metrics beyond simple coverage.

In the presented case study, the DTN-enabled decentralized solution provides 21% more value over 6 hours than the centralized implementation of the same algorithm. If we lower the transiency of the phenomena to an hour (currently 15 mins for precipitation) i.e. time resolution of OSSE-outputs; or if we focus on the poles (currently mid-latitude floods) where there is more FOR overlap i.e. increased processing complexity, and ISL interference i.e. more DTN latency, the centralized solution may provide more value. The proposed scheduler may be evaluated for a given user scenario, and run either way or as a combination.

The time taken to run the algorithm per sat was 1% of the planning horizon, evaluated on MATLAB installed in a Mac OS X v10.13.6 with a 2.6 GHz processor and 16 GB of 2400 MHz memory.

## Acknowledgements

Funded by the NASA New Investigator Program, Earth Science Technology Office, and the Interplanetary Network Directorate at the Jet Propulsion Laboratory.

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
