# OpenReview forum: "Autonomous Scheduling of Agile Spacecraft Constellations with Delay Tolerant Networking for Reactive Imaging"
_icaps-conference.org/ICAPS/2019/Workshop/SPARK — SPARK 2019_

### Official Review · AnonReviewer1 · 2019-05-04
**Comments**

**Rating:** 4
**Confidence:** 1

**Review:**

 Quality & Significance:
The paper motivates the problem statement really well and explanation for major contributions are laid out precisely. The primary results of this work are promising. The work still needs to address scalability, redundancy and  reliability concerns surrounding DTN.

Clarity:
The paper captures details regarding scheduler and DTN is quite depth and with some clarity. But certain sections in the paper (Methodology, Nature Run for Observing System Simulations, Orbital Mechanics and Attitude Control) are slightly tedious to follow and diverts attention from crux of the paper.

Originality:
The paper shows interesting contribution in terms on delivering on-board scheduler without significant loss in % of all GP observed. It also shows applicability of Delay/Disruption Tolerant Networking (DTN) for near-Earth orbiting spacecraft.

Questions for authors:
1. Is it unclear how "Cumulative Value" in Table 1 is calculated?

2. Similarly the paper claims, "The DTN-enabled decentralized solution provides 16% more value over 6 hours than the centralized implementation of the same algorithm.", the exact experimentation showing that is not reflected or clear.

3. The impact of <4 deg FOV constraint on applicability is unclear and can benefit from discussion.

4. As pointed by authors, on-board scheduler is heavily reliant of DTN's performance, even though DTN has shown the time to reach a region for by satellites with priority>3 is long enough for all bundles to be delivered
for perfect consensus, deeper exploration in DTN's performance in terms of size and shape constellations would be interesting.

Few minor fixes:
1. Figure reference issues:
	a. Figure 1 mention in methodology section should be Figure 2.
	b. Figure 3 reference in "Nature run for Observing system simulation" should be Figure 4.

2. Typo on page 2, 2nd paragraph: The algorithm is "mow" broadened in application scope

3. Extra period in the last line above "Methodology" section.

---

### Official Review · AnonReviewer2 · 2019-05-04
**Important and good results, but difficult to read**

**Rating:** 3
**Confidence:** 2

**Review:**

I am not familiar with the problem of spacecraft constellations, however, it looks like an important problem, and the paper proposed a communication module to optimize the schedule of communication. The authors compare the algorithm in two settings: distributed one and centralized one. The results show the distributed approach can achieve 16% more values than the centralized one, with a price of 4% less GP observed.

The presentation of the paper can be improved. I understand that it is an application paper. Nevertheless, the scheduling algorithm  and the results should be described in more detail. The subsections of Nature Run for Observing Simulations and Orbital Mechanics and Attitude Control are rather long, and readers could get lost in those details without right background. I suggest to shorten these sections, but add more details in the algorithm.  Given the current descriptions, I don't understand how the DP based algorithm works. The summarized algorithm needs to be elaborated. For instance, what are commBundles(), tSrc_c, satsWoverlappingFOR(sat)? It will be very helpful if authors can add some descriptions for each for loop in the algorithm to state what loops are doing.  In addition, authors mentioned the DP algorithm is validated by an IP model. The IP model is not formulated in the paper, and there are no details about experiments. How many instances were run? how large are the instances?  and, can the proposed algorithm handle instances with overlapping areas?

In the result section, it is not explained what is cumulative value. In addition, the robustness of the algorithm is not discussed. It is not clear what the performance would be for different instances. It will be good if authors can add some discussions there.

---

### Public Comment · ~Christophe_GUETTIER1 · 2019-05-01
**General Comment**

The combination of DTN and spacecraft agent forming a constellation is an interesting idea. As proposed by authors, the critical point for the integration of both is certainly the scheduling problem. The state of the art is relevant but in general DTN work in a very asynchronous way, without single point of failure. A scheduling technique is also proposed, but many other design options could be taken and they could be discussed.

One thing I did not understand, is whether the model is centralized or the author develop a form of distributed agreement such as consensus (as pointed out in figure 5). Even under a centralized scheduling approach, a form of leader election (and therefore distributed agreement) would be necessary...

---

> ### Public Comment · ~Sreeja_Nag1 · 2019-05-02
> **Response to Christophe Guettier**
>
> The algorithm is flexible in its implementation - can run onboard satellites in a decentralized way or on the ground in a centralized way (pg 2 col 2). It does not implement or optimize for consensus. Instead, it enables every satellite to use the most current information it has (continuously updated by the DTN connections) and make decisions accordingly. So every satellite's decisions depend on its knowledge of previous satellites' implemented actions. The application use case is too rapid for consensus iterations.

---

### Decision · Program_Chairs · 2019-05-08
**Acceptance Decision**

**Decision:**

Accept

**Comment:**

Worthwhile, could be easier to read